# A Causal Foundation Model for Structure and Outcome Prediction

**Max Zhu** [1]  **Martino Mansoldo** [2]  **Ching-Hao Wang** [2]  **Stefan Groha** [2]

## Abstract

We introduce TabPFN-CFM, a causal foundation model that can handle multiple causal problems. TabPFN-CFM predicts both causal structure and outcomes from observational data, supports queries on all three levels of Pearl's Causal Hierarchy and uses known graph structure when available to improve predictions. TabPFN-CFM is trained on synthetic datasets, and generalises to real datasets, demonstrating improved performance over both structural and outcome prediction baselines.

## 1. Introduction

Causal relationships between processes determine how intervening on a system impacts its behaviour. However, determining the causal structure and causal outcomes from observational data is a challenging problem (Pearl, 2000; Imbens & Rubin, 2015). Recently, machine learning models have been developed to make causal predictions on the outcome of interventions efficiently with minimal data (Künzel et al., 2019; Hollmann et al., 2023; Robertson et al., 2025; Balazadeh et al., 2025), as well as structural predictions (Lorch et al., 2022; Ke et al., 2023). We train a causal foundation model that predicts both causal structure and outcomes from observational data. Compared to existing methods, TabPFN-Causal Foundation Model (TabPFN-CFM) improves accuracy, supports all three levels of Pearl's Causal Hierarchy (Pearl, 2000), and leverages known graph structure when available. This is further accompanied by a refined training procedure that improves training efficiency almost 4 times.

### 1.1. Problem Setup

We assume the underlying system follows an SCM $\psi = \{U, V, F\}$, consisting of unobserved variables $U$, observed variables $V$, and unknown structural equations $F$. Unknown variables include unobserved covariates and sources

[1]University of Cambridge, United Kingdom [2]GSK.ai. Correspondence to: Max Zhu <mz406@cam.ac.uk>.

*Proceedings of the $43^{rd}$ ICML Workshop on Foundation Models for Structured Data*, Seoul, South Korea. PMLR 306, 2026.

of noise. The observed variables $V$ are split into covariates $X$, a binary treatment $T$ and an outcome $Y$. The structural equations $F$ define each variable's parents and form a graph $\mathcal{G}$. We have access to an observational dataset $\mathcal{D}^{\mathrm{obs}} = \{\mathbf{x}_i, y_i, t_i\}_{i=1}^n$, consisting of $n$ i.i.d. samples drawn from $\psi$, and optionally, prior knowledge of the causal graph $\mathcal{G}^{\mathrm{est}}$. If there is no prior then $\mathcal{G}^{\mathrm{est}} = \varnothing$.

Our model targets two objectives. Firstly, if the true causal graph is unknown, our model can make an estimate of the underlying causal graph structure. Second is outcome predictiton under three types of causal queries. Given a samples covariates $\mathbf{x}^*$, natural treatment $t^*$ and outcome $y^*$, the Observational Query, $P(y^*|\mathbf{x}^*, T = t^*, \mathcal{D}^{\mathrm{obs}}, \mathcal{G}^{\mathrm{est}})$, predicts the outcome when the treatment takes its observed value. The Interventional Query, $P(y^*|\mathbf{x}^*, \mathrm{do}(T = 1 - t^*), \mathcal{D}^{\mathrm{obs}}, \mathcal{G}^{\mathrm{est}})$, predicts the outcome under an externally assigned treatment which differs from the observed treatment. The Counterfactual Query, $P(y^*|\mathbf{x}^*, \mathrm{do}(T = 1 - t^*), y = y_{t^*}, \mathcal{D}^{\mathrm{obs}}, \mathcal{G}^{\mathrm{est}})$, predicts the what-if outcome under a different treatment, conditional on the outcome observed under the original treatment. By training a single model jointly on all tasks, the model gains a broader understanding of causal processes and learns more efficiently than models trained on a single task.

oTo approach this problem, we follow the Bayesian PFN framework (Robertson et al., 2025; Balazadeh et al., 2025). A prior over SCMs, $p(\psi)$, generates a true SCM, $\psi_{\mathrm{true}} \sim p(\psi)$. Observational data yields the posterior, $P(\psi|\mathcal{D}^{\mathrm{obs}}, \mathcal{G}^{\mathrm{est}})$. Given an SCM, the outcome can be estimated as $P(y|\mathbf{x}^*, \mathrm{do}(T = t^*), \psi)$. The target distribution for a causal query, say an Interventional Query, is

$$P(y^*|\mathbf{x}^*, \mathrm{do}(T = 1 - t^*), \mathcal{D}^{\mathrm{obs}}, \mathcal{G}^{\mathrm{est}}) = \tag{1}$$

$$\int P(y^*|\mathbf{x}^*, \mathrm{do}(T = 1 - t^*), \psi) P(\psi|\mathcal{D}^{\mathrm{obs}}, \mathcal{G}^{\mathrm{est}}) d\psi. \tag{2}$$

If the graph structure $\mathcal{G}$ is unknown, its posterior is

$$P(\mathcal{G}^*|\mathcal{D}^{\mathrm{obs}}) = \int P(\mathcal{G}^*|\psi) P(\psi|\mathcal{D}^{\mathrm{obs}}) d\psi. \tag{3}$$

The optional graph allows the posterior to be estimated more accurately, by providing information about the causal relationships between variables. In Appendix A, we prove that including the graph as input can only improve posterior

accuracy, with no improvement occurring if the graph does not change the posterior distribution of $\psi$ (where $y$ has support).

In practice, instead of explicitly estimating $P(\psi|\mathcal{D}^{\mathrm{obs}})$, we train a model to directly estimate $P(y^*|\mathbf{x}^*, \mathrm{do}(T = t^*), \mathcal{D}^{\mathrm{obs}})$ and $P(\mathcal{G}|\mathcal{D}^{\mathrm{obs}})$, by drawing samples from the prior $p(\psi)$ and optimising log likelihoods conditioned on observations, $L = -E[\log \hat{p}_\theta(y^*|\mathbf{x}^*, \mathrm{do}(T = 1 - t^*), \mathcal{D}^{\mathrm{obs}}, \mathcal{G})]$. We show this loss is equivalent to optimising the KL divergence between the model's estimated distribution and the true posterior distribution in Appendix B.

Existing deep learning methods for causal structure learning do not predict confounding variables caused by unobserved factors. Our method addresses this by using Acyclic Directed Mixed Graphs (ADMGs), which represent unobserved confounding via bidirected edges; details are provided in Appendix C. Confounding may lead to unidentifiable graphs, so $P(\psi|\mathcal{D}^{\mathrm{obs}}, \mathcal{G}^{\mathrm{est}})$ is uncertain, which is reflected in the estimated distributions. By predicting confounding, our model not only sheds light on the underlying causal structure, but also allows the user to account for confounding when interpreting results.

## 2. Data generation

Our model is trained on synthetic data generated from a prior distribution of SCMs to approximate Bayesian inference on this prior. First, we sample SCMs with random graph structures, missing nodes, noise distributions, and structural equations. Random functions are generated using neural networks with random weights and nonlinearities, along with random noise distributions.

To generate a single training sample, for each SCM, we sample observational data $\mathcal{D}^{\mathrm{obs}}$ from the SCM and generate another observational datapoint for prediction, $\mathcal{D}^{\mathrm{pred}} = \{\mathbf{x}^*, t^*, y^*\}$. Since the true causal graph is known, we generate a counterfactual (with fixed noise), $\mathcal{D}^{\mathrm{causal}} = \{\mathbf{x}^*_t, t = \mathrm{do}(1 - t^*), y^*_t\}$. Finally, we have the ADMG corresponding to the SCM structure, $\mathcal{G}^{\mathrm{est}} = \{A, C\}$, with adjacency matrix $A$ and bidirected confounding matrix $C$.

Our setup extends the priors used in Robertson et al. (2025). This diversity in the prior ensures our model learns to perform inference on a wide variety of causal systems, improving its generalization to real data. A full description of the data generation process is given in Appendix D.

## 3. Model architecture

### 3.1. Architecture overview

Our model extends the Do-PFN architecture to support interventional and counterfactual prediction with explicit graph

conditioning. Details are given in Appendix E.

The encoder embeds the fit set $\mathcal{D}^{\mathrm{fit}}$, the prediction query $\mathcal{D}^{\mathrm{pred}}$, and the prior ADMG structure. Covariates, outcomes and treatments are embedded with randomized column positional embeddings as in TabPFNv2, along with the query type. The ADMG is encoded through the adjacency $A$ and bidirected matrix $C$ as well as the derived ancestral matrix, giving the model direct access to parent, child, ancestor, and descendant relationship (Ke et al., 2023).

The main transformer consists of row-wise attention, column-wise attention and feed-forward layers. The predictions additionally attend to the graph embeddings, enabling model predictions to incorporate the graph prior.

The outputs are computed from the final hidden state. The final hidden state corresponding to the outcome variable is passed through an MLP head to produce logits over discretized outcome buckets, yielding the predictive distribution $\hat{p}_\theta(y)$. The graph structure predictions are generated using the final hidden states corresponding to each feature variable. A decoder computes elementwise predictions for every possible edge in the graph. Predictions are made for the directed adjacency matrix $\hat{A}$, the bidirected correlation matrix $\hat{C}$ and the ancestral matrix $\hat{R}$.

In our architecture, attention flows from $\mathcal{D}^{\mathrm{fit}}$ and the graph to $\mathcal{D}^{\mathrm{pred}}$, but not the other way around. In addition to allowing efficient inference from the KV cache, as in TabPFN, this makes the predicted graph structure independent of the prediction query, including the input graph prior.

Compared with TabPFNv2 and Do-PFN, we omit feature grouping, ensembling, random augmentations, and random feature products because the graph explicitly indexes the variables and would be difficult to preserve under feature mixing. Furthermore, we also introduce several training and backbone improvements for stability and efficiency which together improve training speed 3-4x, with significantly lower final loss. These changes are evaluated in the ablation study in Appendix F.

### 3.2. Training procedure

Each training datapoint consists of datasets $\mathcal{D}^{\mathrm{fit}}$, $\mathcal{D}^{\mathrm{pred}} = \{\mathbf{x}^*, t^*, y^*\}$, $\mathcal{D}^{\mathrm{causal}} = \{\mathbf{x}^*_t, t = do(1 - t^*), y^*_t\}$, and $\mathcal{G}^{\mathrm{est}}$. Each datapoint is used to generate samples for all three query types. In the observational query the model is given $\mathbf{x}^*$ and $t^*$, and the target is $y^*$. In the interventional query the model is given $\mathbf{x}^*$ and $do(1 - t^*)$, and the target is $y^*_t$. In the counterfactual query, the model is given $\mathbf{x}^*_t$, $do(1 - t^*)$ and $y^*$, and the target is $y^*_t$. In each case, let $y_{\mathrm{targ}}$ and $\hat{p}_\theta(y)$ be the target and predicted distribution, respectively. The prediction loss is

$$\mathcal{L}_{\mathrm{pred}} = CE(y_{\mathrm{targ}}, \hat{p}_\theta(y)). \tag{4}$$

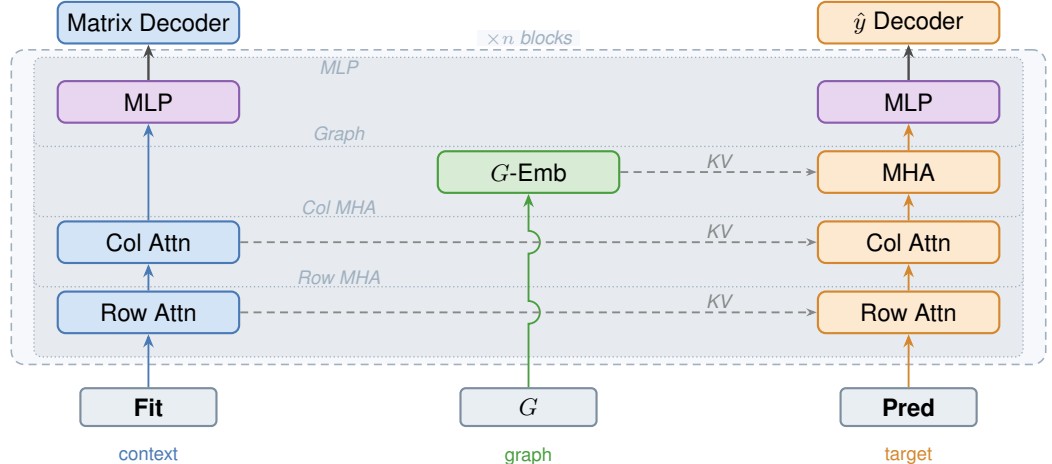

*Figure 1.* Model architecture diagram.

This is the standard prediction loss for PFNs (Hollmann et al., 2023; Balazadeh et al., 2025; Robertson et al., 2025). The model is always given $\mathcal{D}^{\text{fit}}$. In order to allow the model to learn to make predictions with and without the true causal graph, $\mathcal{G}^{\text{est}}$ is set to zero half the time.

The structural prediction losses between predictions $\hat{A}, \hat{R}, \hat{C}$ and true matrices are the elementwise binary cross entropy,

$$\mathcal{L}_{\text{graph}} = \frac{1}{(k+2)^2} \sum_{i=1}^{k+2} \sum_{j=1}^{k+2} BCE(M_{i,j}, \hat{M}_{i,j}), \quad (5)$$

$$M \in \{A, R, C\}, \hat{M} \in \{\hat{A}, \hat{R}, \hat{C}\}. \quad (6)$$

Since the structural predictions are independent of the query, this loss is computed only once per datapoint. The overall loss is

$$\mathcal{L} = \mathcal{L}_{\text{pred}} + \lambda \cdot \mathcal{L}_{\text{graph}}. \quad (7)$$

Finally, the model is trained with dummy features from batch padding. Although samples are processed independently, the model can attend to these dummy features; empirically, adding them at inference improves performance, suggesting they may support intermediate computations.

## 4. Evaluations

### 4.1. Synthetic toy examples

First, we showcase TabPFN-CFM on the Instrumental Variable (IV) problem with SEM: $Z \to T \to Y, U \to T, U \to Y$. A linear IV SEM is generated with known parameters and noise distributions, which allows the exact observational, interventional and counterfactual distributions to be computed. A sample is drawn from this SEM, and we evaluate the model predictions on this sample, both with and without the graph structure as input. See Appendix G.1 for

details. Figures 2 and 5 show the predicted and exact distributions for the three query types, without and with the prior structure respectively. The model predictions align well in all cases.

Table 3 shows the predicted adjacency matrix. The predicted adjacency matrix closely matches the true adjacency matrix, indicating that the model has effectively learned the underlying causal structure of the SEM.

A more complex nonlinear SEM is investigated in Appendix G.2 and Figure 7 shows how predictions improve as the number of fit samples increases.

### 4.2. Synthetic SEMs

Our model is evaluated on a large number of synthetic SEMs drawn from our prior dataset distribution as well as out of distribution (OOD) priors. For consistency, 500 datasets are drawn and saved for each setting used for evaluation. The OOD dataset is generated with random Fourier functions as nonlinearities instead of neural networks, see Appendix H. This tests our model's ability to generalise to unseen functional forms. The model is compared both with and without the true graph structure as input.

We evaluate the model's prediction mean squared error (MSE) for observational, interventional and counterfactual queries, and accuracy and area under the receiver operating characteristic curve (AUC) of the graph structure prediction.

For outcome prediction, we compare against Do-PFN and established meta-learners, the S-learner, T-learner, X-learner and Doubly Robust (DR) learner, all using TabPFNv2.5 as the base model (Künzel et al., 2019; Battocchi et al., 2019). The baseline learners are unable to make counterfactual predictions. For graph prediction, we compare against AVICI (Lorch et al., 2022), a strong deep-learning graph prediction

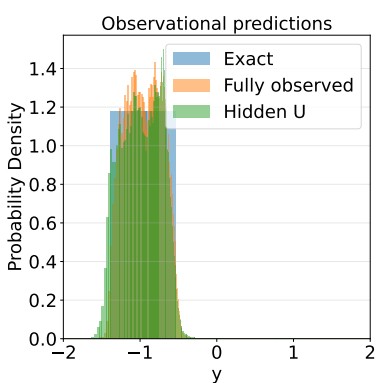 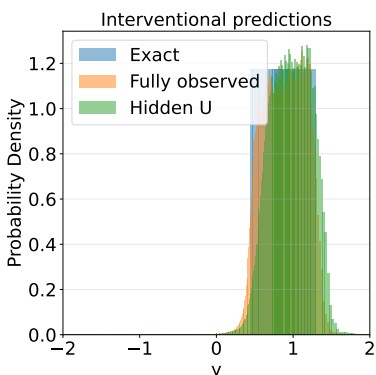 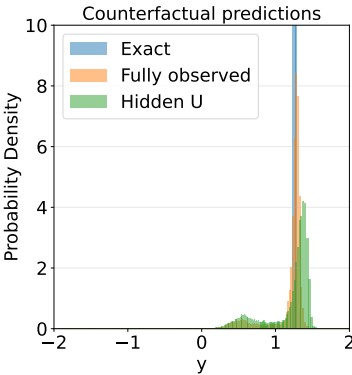

*Figure 2.* Observational (left), Interventional (center), and Counterfactual (right) distributions for the IV example. Exact solutions are in blue, model predictions in orange for when $U$ is observed, and in green when $U$ is unobserved. Model is not given the graph structure.

*Table 1.* Predictions on OOD SEMs with $n = 1024$ fit samples, compared by mean squared error ($\downarrow$). The baseline methods are unable to make counterfactual predictions.

| Model | Obs. | Int. | Count. |
|---|---|---|---|
| S-learner | 0.565 | 0.888 | - |
| T-learner | 0.574 | 0.909 | - |
| X-learner | 0.574 | 0.909 | - |
| DR-Learner | 0.577 | 0.885 | - |
| Do-PFN | 0.782 | 0.966 | - |
| TabPFN-CFM | **0.537** | 0.765 | 0.607 |
| TabPFN-CFM W/Graph | 0.538 | **0.739** | **0.572** |

*Table 2.* Structural predictions on synthetic OOD SEMs with $n = 1024$ fit samples, evaluated on predicted adjacency, ancestral and confounding matrices, averaged over the dataset. Main values tracks the average AUROC ($\uparrow$), with accuracy (%)($\uparrow$) in brackets.

| Model | Adjacency | Ancestral | Confounding |
|---|---|---|---|
| FCI | 0.533 (88) | 0.523 (81) | 0.563 (87) |
| GES | 0.654 (81) | 0.654 (60) | - |
| LiNGAM | 0.558 (78) | 0.524 (70) | - |
| PC | 0.685 (86) | 0.639 (55) | - |
| AVICI | 0.623 (89) | 0.596 (82) | - |
| TabPFN-CFM | **0.889 (91)** | **0.902 (88)** | **0.828 (93)** |

model as well as established methods FCI (Spirtes et al., 1995), GES (Chickering, 2002), LiNGAM (Shimizu et al., 2006), and PC (Spirtes et al., 2000). Only FCI can predict confounding.

Tables 12 and 1 summarise outcome prediction results for in-distribution and out-of-distribution SEMs, respectively. Our model generalises well to out of distribution SEMs, indicating it is not overfit to our prior. In the observational setting, TabPFN-CFM is comparable to the baselines, likely due to the strong TabPFNv2.5 base model used by the Meta-Learners. Our model outperforms all the baselines on the interventional setting, even without the graph structure. Giving the model the true graph structure improves predictions in the causal interventional and counterfactual settings but not the observational setting, as causal direction does not affect the observational distribution.

Tables 13 and 2 summarise the structure prediction results where our model outperforms the baselines. As well as directly comparing the predicted graphs, we also use the predicted adjacency matrix to generate an ancestral matrix using monte-carlo sampling in Table 8. This is significantly less accurate than directly using the predicted ancestral matrix, showing the advantage of directly predicting the ances-

tral structure rather than relying on the adjacency matrix. Full results with varying fit samples are shown in Table 14, 15, 16 and 17.

### 4.3. Real Data

Our model is evaluated on two real datasets, the Amazon Sales dataset (Blöbaum et al., 2024) and the Law School Admissions dataset (Wightman, 1998). These datasets have established causal graphs and counterfactuals generated with the DoWhy framework, taken from Robertson et al. (2025). The results are shown in Table 4 and Table 5. While all models are similar in their observational performance, our model excels in the interventional settings. In the Law School Admissions dataset, our model is able to use the counterfactual information to significantly improve its accuracy.

### 5. Discussion

We have introduced TabPFN-CFM, a causal foundation model that handles multiple causal tasks, including both causal structure and outcome prediction. We believe causal PFNs can continue to be extended to more complex settings, such as time series and cyclic causal settings.

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

*Table 3.* Predicted adjacency matrices for the linear SEM experiment on the IV graph. Values represent predicted edge probabilities. The true edges are $Z \to T, T \to Y, U \to T$ and $U \to Y$.

|   | T | Z | U | Y |
|---|---|---|---|---|
| T | - | 0.078 | 0.010 | 0.997 |
| Z | 0.873 | - | 0.208 | 0.129 |
| U | 0.780 | 0.460 | - | 0.978 |
| Y | 0.000 | 0.006 | 0.002 | - |

*Table 4.* Comparison of our model with baselines on the Amazon Sales dataset, compared by mean squared error ($\downarrow$). The baseline methods are unable to use the true causal graph or make counterfactual predictions.

| Model | Obs. | Int. | Count. |
|---|---|---|---|
| S-learner | 0.002 | 7.269 | - |
| T-learner | **0.001** | 0.964 | - |
| X-learner | **0.001** | 0.964 | - |
| DR-Learner | 0.011 | 8.716 | - |
| Do-PFN | 0.006 | 0.593 | - |
| TabPFN-CFM | 0.013 | **0.455** | 0.505 |
| TabPFN-CFM W/Graph | 0.012 | 0.463 | **0.502** |

*Table 5.* Comparison of our model with baselines on the Law School Admissions dataset, compared by mean squared error ($\downarrow$). The baseline methods are unable to make counterfactual predictions.

| Model | Obs. | Int. | Count. |
|---|---|---|---|
| S-learner | 0.871 | 0.931 | - |
| T-learner | **0.870** | 0.928 | - |
| X-learner | **0.870** | 0.928 | - |
| DR-Learner | 0.872 | 0.934 | - |
| Do-PFN | 0.893 | 0.900 | - |
| TabPFN-CFM | 0.873 | 0.904 | **0.124** |
| TabPFN-CFM W/Graph | 0.872 | **0.896** | 0.145 |

## A. Comparing estimation with and without graph input

Including the graph input can improve model predictions. For simplicity, ignore $c$ here. With and without $\mathcal{G}$, the target distributions are

$$p_0(y|D^{\mathrm{obs}}) = \int p(y|\psi)p(\psi|D^{\mathrm{obs}}) \, d\psi \tag{8}$$

$$p_G(y|D^{\mathrm{obs}}, \mathcal{G}) = \int p(y|\psi)p(\psi|D^{\mathrm{obs}}, \mathcal{G}) \, d\psi \tag{9}$$

$$\tag{10}$$

Comparing their losses, we have:

$$E_{y|D^{\mathrm{obs}}, \mathcal{G}} \left[ -\log p_G(y|D^{\mathrm{obs}}, \mathcal{G}) - (-\log p_0(y|D^{\mathrm{obs}})) \right] = \int -p_G(y|D^{\mathrm{obs}}, \mathcal{G}) \log \frac{p_G(y|D^{\mathrm{obs}}, \mathcal{G})}{p_0(y|D^{\mathrm{obs}})} \, dy \tag{11}$$

$$= -KL(p_G(y|D^{\mathrm{obs}}, \mathcal{G})||p_0(y|D^{\mathrm{obs}})) \leq 0 \tag{12}$$

*Table 6.* Structural predictions on the Amazon Sales dataset, evaluated on predicted adjacency matrix and ancestral matrix. Main values tracks the AUROC (↑), with accuracy (↑)in brackets.

| Model | Adjacency | Ancestral |
|---|---|---|
| FCI | 0.523 (0.79) | 0.532 (0.63) |
| GES | 0.527 (0.53) | 0.603 (0.60) |
| LiNGAM | 0.464 (0.59) | 0.600 (0.62) |
| PC | 0.573 (0.72) | 0.555 (0.57) |
| AVICI | 0.698 (0.78) | **0.872 (0.81)** |
| TabPFN-CFM | **0.708 (0.84)** | 0.701 (0.72) |

*Table 7.* Structural predictions on the Law School Admissions dataset, evaluated on predicted adjacency matrix and ancestral matrix. Main values tracks the AUROC (↑), with accuracy (↑)in brackets.

| Model | Adjacency | Ancestral |
|---|---|---|
| FCI | 0.500 (0.56) | 0.500 (0.63) |
| GES | 0.627 (0.56) | 0.700 (0.63) |
| LiNGAM | 0.353 (0.43) | 0.440 (0.48) |
| PC | 0.671 (0.56) | 0.700 (0.63) |
| AVICI | 0.545 (0.69) | 0.517 (0.63) |
| TabPFN-CFM | **0.886 (0.93)** | **0.912 (0.88)** |

*Table 8.* Comparison of direct ancestral matrix prediction and ancestral matrix generated from the predicted adjacency matrix using Monte-Carlo sampling for out-of-distribution SEMs, across different numbers of fit data points (n-fit). Values represent AUC (↑).

| n-fit | Direct ancestral | Ancestral from adjacency |
|---|---|---|
| 32 | 0.712 | 0.530 |
| 256 | 0.848 | 0.616 |
| 1024 | 0.902 | 0.693 |

Now, taking expectation over $D^{\mathrm{obs}}$ and $\mathcal{G}$, we have

$$L_G - L_0 = E_{y,D^{\mathrm{obs}},\mathcal{G}}\left[-\log p_G(y|D^{\mathrm{obs}},\mathcal{G}) - (-\log p_0(y|D^{\mathrm{obs}}))\right] \tag{13}$$

$$= -E_{y,D^{\mathrm{obs}},\mathcal{G}}\left[KL(p_G(y|D^{\mathrm{obs}},\mathcal{G})||p_0(y|D^{\mathrm{obs}}))\right] \leq 0 \tag{14}$$

Therefore, including the graph input can only improve the model predictions, with no improvement occurring if $p_G(y|D^{\mathrm{obs}},\mathcal{G}) = p_0(y|D^{\mathrm{obs}})$. Equivalently, in terms of the posterior of $\psi$, there is an improvement if

$$\int p(y|\psi)p(\psi|D^{\mathrm{obs}},\mathcal{G})\,d\psi \neq \int p(y|\psi)p(\psi|D^{\mathrm{obs}})\,d\psi \tag{15}$$

$$\int p(y|\psi)\left[p(\psi|D^{\mathrm{obs}},\mathcal{G}) - p(\psi|D^{\mathrm{obs}})\right]\,d\psi \neq 0 \tag{16}$$

$$p(\psi|D^{\mathrm{obs}},\mathcal{G}) \neq p(\psi|D^{\mathrm{obs}}) \tag{17}$$

$$\tag{18}$$

So an improvement occurs if including $\mathcal{G}$ changes the posterior distribution of $\psi$ (where $y$ has support). The notable case where this does not occur is when the graph structure is uniquely identified by the observational data, $p(\psi|D^{\mathrm{obs}})$ is a delta distribution on the true SCM, and $p(\psi|D^{\mathrm{obs}},\mathcal{G})$ is the same delta distribution, the SEM is identifiable. In the finite data regime, even if the SEM is identifiable, the posterior may not be perfect, so including the graph input can still improve the model predictions.

# B. Appendix: Proof of equivalence between log probability and KL divergence

Proof minimising log probability (and cross-entropy) equals minimising KL divergence. This proof extends the proof given in TabPFN (Hollmann et al., 2023) and Do-PFN (Robertson et al., 2025) by including the conditioning on the graph input. We have the following definitions and properties:

$$\psi \sim p(\psi) \tag{19}$$
$$D^{\mathrm{obs}} \sim p(D^{\mathrm{obs}}|\psi) \tag{20}$$
$$\mathcal{G} \sim p(\mathcal{G}|\psi) \tag{21}$$
$$c \sim p(c|\psi) \tag{22}$$
$$y \sim p(y|c,\psi) \tag{23}$$
$$L = -E[\log \hat{p}_\theta(y|c,D^{\mathrm{obs}},\mathcal{G})] \tag{24}$$
$$\{c,y\} \perp\!\!\!\perp D^{\mathrm{obs}}|\psi \tag{25}$$
$$\{c,y\} \perp\!\!\!\perp \mathcal{G}|\psi \tag{26}$$
$$\tag{27}$$

Optimising the loss function $L$ is equivalent to optimising the KL divergence between the true distribution and the model distribution. We have,

$$L = -E[\log \hat{p}_\theta(y|c,D^{\mathrm{obs}},\mathcal{G})] \tag{28}$$

$$= -\int\int\int \log \hat{p}_\theta(y|c,D^{\mathrm{obs}},\mathcal{G})p(y,c,D^{\mathrm{obs}},\mathcal{G})\,dy\,dc\,dD^{\mathrm{obs}}d\mathcal{G} \tag{29}$$

$$= -\int\int\int \log \hat{p}_\theta(y|c,D^{\mathrm{obs}},\mathcal{G})p(y|c,D^{\mathrm{obs}},\mathcal{G})p(c,D^{\mathrm{obs}},\mathcal{G})\,dy\,dc\,dD^{\mathrm{obs}}d\mathcal{G} \tag{30}$$

$$= -E_{c,D^{\mathrm{obs}},\mathcal{G}}\left[\int p(y|c,D^{\mathrm{obs}},\mathcal{G})\log \hat{p}_\theta(y|c,D^{\mathrm{obs}},\mathcal{G})\,dy\right] \tag{31}$$

$$= -E_{c,D^{\mathrm{obs}},\mathcal{G}}\left[\int p(y|c,D^{\mathrm{obs}},\mathcal{G})\left[\log\frac{\hat{p}_\theta(y|c,D^{\mathrm{obs}},\mathcal{G})}{p(y|c,D^{\mathrm{obs}},\mathcal{G})} + \log p(y|c,D^{\mathrm{obs}},\mathcal{G})\right]\,dy\right] \tag{32}$$

$$= E_{c,D^{\mathrm{obs}},\mathcal{G}}\left[KL(p(y|c,D^{\mathrm{obs}},\mathcal{G})||\hat{p}_\theta(y|c,D^{\mathrm{obs}},\mathcal{G})) + C)\right] \tag{33}$$

$$\tag{34}$$

hence minimising the log probability is equivalent to minimising the KL divergence between the true distribution and the model distribution. This can also be written with $\psi$ more explicitly. Using the independence property, we have

$$p(y|c, D^{\text{obs}}, \mathcal{G}) = \int p(y|c, D^{\text{obs}}, \mathcal{G}, \psi) p(\psi|c, D^{\text{obs}}, \mathcal{G}) \, d\psi \tag{35}$$

$$= \int p(y|c, \psi) p(\psi|c, D^{\text{obs}}, \mathcal{G}) \, d\psi. \tag{36}$$

$$\tag{37}$$

Hence,

$$L = -E_{c, D^{\text{obs}}, \mathcal{G}} \left[ \int p(y|c, D^{\text{obs}}, \mathcal{G}) \log \hat{p}_\theta(y|c, D^{\text{obs}}, \mathcal{G}) \, dy \right] \tag{38}$$

$$= -E_{c, D^{\text{obs}}, \mathcal{G}} \left[ \int \int p(y|c, \psi) p(\psi|c, D^{\text{obs}}, \mathcal{G}) \log \hat{p}_\theta(y|c, D^{\text{obs}}, \mathcal{G}) \, dy \, d\psi \right] \tag{39}$$

$$= -E_{c, D^{\text{obs}}, \mathcal{G}, \psi} \left[ \int p(y|c, \psi) \log \hat{p}_\theta(y|c, D^{\text{obs}}, \mathcal{G}) \, dy \right] \tag{40}$$

$$= E_{c, D^{\text{obs}}, \mathcal{G}, \psi} \left[ KL(p(y|c, \psi) || \hat{p}_\theta(y|c, D^{\text{obs}}, \mathcal{G})) + C) \right] \tag{41}$$

$$\tag{42}$$

## C. Causal Graphs and Structural Causal Models

A *nonparametric structural equation model with independent errors* (NPSEM-IE) satisfies the following conditions (Pearl, 2000):

1. For each $i = 1, \ldots, d$, there exists a set of parent variables $\text{Pa}_i \subseteq V \setminus \{V_i\}$ such that

$$V_i = f_i(\text{Pa}_i, U_i).$$

2. The graph with vertex set $V$ and directed edges $V_j \to V_i$ whenever $V_j \in \text{Pa}_i$ is acyclic.

3. The unobserved variables are mutually independent and exogenous:

$$U_i \perp\!\!\!\perp U_j \quad \text{for } i \neq j.$$

If our causal system satisfies an NPSEM-IE (or can be trivially reduced to one), corresponding to only "noise" variables being unobserved, then the causal graph corresponding to observed variables $V$ will be a DAG structure. Each variable is independent of its non-descendants given its parents (the local Markov property), with the direction of edges indicating the direction of causality.

However, in the more general case of a DAG over variables $(U, V)$, the NPSEM-IE assumptions need not hold. The unobserved variables in $U$ may be endogenous (i.e., have parents in $V$ or in $U$), a single unobserved variable may have multiple children in $V$, and the unobserved variables may be statistically dependent. After marginalizing out $U$, the causal relationships among the observed variables $V$ can be represented by an acyclic directed mixed graph (ADMG) (Richardson, 2003), which contains both directed edges (indicating directed causal influence) and bidirected edges (indicating the presence of unobserved common causes, i.e. unobserved confounding). Such cases are common in real-world scenarios, where not all relevant variables can be observed or measured.

There are many models referred to as ADMGs in the literature (Zhao, 2025); here we will refer to ADMGs as DAGs with unobserved variables marginalized out. An ADMG graph consists of an adjacency matrix, $A$, and correlation matrix, $C$. The adjacency matrix contains directed causal relationships, while the symmetric correlation matrix contains bidirected non-causal correlations caused by unobserved confounding.

Finally, we note there is a more general case, where the underlying structure is not a DAG and instead contains cycles. Such structures can arise in systems with temporal feedback loops or cyclic causality. However, in this work we focus on the acyclic case (DAGs and ADMGs) and leave cyclic structures for future work.

# D. Dataset generation

## D.1. Generating random SCMs

Firstly, the DAG structure is generated. The DAG size, average connectivity degree and type of DAG are sampled. The type of DAG is either Erdos-Renyi or Scale-Free. For Scale-Free graphs, the attractiveness parameter is also sampled. Given these parameters, a random DAG is generated. A treatment node and target node are sampled, with bias towards selecting the treatment as a causal ancestor of the target to ensure meaningful interventions.

The structural equations are sampled for every node in the DAG. First, the nonlinearity is sampled from a set of possible functions, including whether to use an MLP (Table 9). Then, the type of noise, position of noise and noise distribution are sampled (Table 10). Relevant distribution parameters (e.g. variance) are also sampled randomly. Finally, the node's weight vector $\mathbf{W}$ and bias scalar $b$ are drawn from the Xavier Normal distribution (Glorot & Bengio, 2010).

The node's structural equation takes as input the concatenation of its parent values, $\mathbf{x}_{\text{pa}}$, and computes the node's value in the most general form as:

$$x_{\text{out}} = g(\mathbf{W}_2 f_{\text{nonlinear}}(\mathbf{W}_1 \cdot g(\mathbf{x}_{\text{pa}}, \epsilon_{\text{pre}}) + b), \epsilon_{\text{post}}) \tag{43}$$

Where $f$ is the elementwise nonlinearity, $g$ is the noise composition function (either elementwise additive or multiplicative), $\epsilon_{\text{pre}}$ and $\epsilon_{\text{post}}$ are noise variables sampled from the pre and post noise distribution. If the node has no parents, $\mathbf{x}_{\text{pa}}$ is empty and only noise is used as input to the node.

The treatment node is generated differently, since it is a binary variable. The treatment node's structural equation is:

$$\tilde{x}_{\text{out}} = \mathbf{W} \cdot (\mathbf{x}_{\text{pa}} + \epsilon_{\text{pre}}) \tag{44}$$

$$x_{\text{out}} = a \cdot \mathbb{1}[\tilde{x}_{\text{out}} > x_{\text{thresh}}] + b \tag{45}$$

where $a, b$ are random scalars and $x_{\text{thresh}}$ is set so the observational distribution of $\tilde{x}_{\text{out}}$ has a randomly sampled class balance. This ensures the treatment always contains positive and negative samples.

Finally, a subset of nodes in the DAG are selected as observed variables $V$, with the remaining nodes treated as unobserved variables $U$. The selection is random, with the constraint that the treatment and target nodes are always observed. Unobserved variables are marginalized out when generating datasets, creating an observed acyclic adjacency matrix, $A$, and bidirected confounding matrix, $C$, between observed nodes together forming the ADMG structure. If the sample is trivial (e.g. all treatments the same or no causal relations), the sample is discarded. This completes the specification of a single SCM, $\psi$. Our synthetic prior distribution of SCMs, $P(\psi)$, is the distribution induced by the above generative process with all parameters sampled randomly.

## D.2. Sampling causal datasets from SCMs

Now, we describe how each sample dataset is generated. A SCM $\psi$ is sampled from the prior $P(\psi)$. Then, for each sample $i$ in the dataset, the exogenous noise variables $U$ are sampled from their respective distributions. The observed variables $V$ are generated by evaluating the structural equations in topological order. This process is repeated $n$ times to generate the observational dataset with $n$ entries, $\mathcal{D}^{\text{fit}} = \{\mathbf{x}_i, t_i, y_i\}_{i=1}^n$. The underlying graph is stored in $G^{\text{est}} = \{A, C\}$.

Then, we generate the prediction dataset. Another sample is drawn from the observational dataset as above, $\mathcal{D}^{\text{pred}} = \{\mathbf{x}^*, t^*, y^*\}$. Since we know the true SCM, the counterfactual sample can also be generated. However, we must be careful to correctly simulate the counterfactual since nodes between $T$ and $Y$ (observed or unobserved) may be descendants of $T$; these nodes should not be fixed. To account for this, we fix every node and noise variable that may be topologically ordered before $T$. That is, we split the nodes into descendants of $T$, $N_{\text{desc}}$, and non-descendants, $N_{\text{nd}}$. All of the noise variables and nodes in $N_{\text{nd}}$ are kept fixed to their sampled values from the observational sample. The treatment node is set to its interventional value $T = \text{do}(1 - t^*)$. Nodes in $N_{\text{desc}}$ are recomputed with interventional $T$. This procedure generates the "what-if" counterfactual outcome. Nodes in $V$ between $T$ and $Y$ will change under intervention, but their counterfactual values will depend on their observed value through fixed parent noise variables, allowing the model to use this information to make more accurate predictions. We denote this new sample as $\mathcal{D}^{\text{causal}} = \{\mathbf{x}_t^*, t = \text{do}(1 - t^*), y_t^*\}$. We note there are alternative methods of generating causal queries, such as fixing only ancestors of $T$ corresponding to interventional rather than counterfactual distribution. These are not used in this paper, but we allow users to easily generate and train new models on these alternative distributions with our codebase by changing a configuration flag.

Our model is trained with up to $n = 1600$ fit samples and $k = 33$ covariates (35 nodes in total).

*Table 9.* Nonliearities. Outputs are scaled to be approxmately standardized. Quantize and Square nonlinearities modified for stability.

| Name | Function |
|---|---|
| Identity | $x$ |
| Relu | $1.67 \cdot \text{relu}(0, x) - 0.688$ |
| Tanh | $1.43 \cdot \tanh(x)$ |
| Sine | $1.43 \cdot \sin(\pi x)$ |
| Quantize (stabilized) | $\lceil x + \epsilon \cdot \text{rand}(0, 1) \rceil$ |
| Square (stabilized) | $5.7 * \log(1 + 1/5.7 * x^2) - 0.83$ |

*Table 10.* Noise generation parameters

| Distribution | Position | Composition |
|---|---|---|
| Normal | Pre-nonlinearity | Multiplicative |
| Uniform | Post-nonlinearity | Additive |
| Bernoulli | Both | |
| Laplace | | |
| Gamma | | |

# E. Model architecture

Our architecture is based on TabPFNv2 and Do-PFN, with several upgrades to handle the additional objectives and improve training efficiency. The model consists of four components, the encoder module, the observation module, the graph module and the prediction module.

## E.1. Encoder module

The encoder module embeds the inputs. For notational simplicity, we define / describe the shapes as follows, $\mathbf{x}^{\text{fit}} \in \mathbb{R}^{n \times k}$, $\mathbf{x}^* \in \mathbb{R}^{1 \times k}$, $t^{\text{fit}} \in \{0, 1\}^n$, $t^* \in \{0, 1\} \times \{0, 1\}$, $y^{\text{fit}} \in \mathbb{R}^n$, $G \in \{0, 1\}^{(k+2) \times (k+2)}$, where $k$ is the number of covariates, and $n$ is the number of observations. The hidden dimension of the transformer is $h$. Both $\mathbf{x}^{\text{fit}}$ and $\mathbf{x}^*$ are encoded with the same linear transform, followed by a column-wise positional embedding:

$$h_a^x = \text{Linear}_x(\mathbf{x} || \text{CDF}(\mathbf{x})) \in \mathbb{R}^{n \times k \times h} \tag{46}$$

$$h^x = h_a^x + \text{PosEmb}_{\text{col}} \in \mathbb{R}^{n \times k \times h}. \tag{47}$$

Each covariate is concatenated with its CDF transform, which helps embedding extreme outliers. Each of the $k$ feature columns of the dataset is given a positional embedding generated by the subspace method from TabPFNv2. For each column $j$

$$\text{PosEmb}_{\text{col } j} = W_{\text{col}} \epsilon \tag{48}$$

$$\epsilon \sim \mathcal{N}(0, I_{h/4}) \tag{49}$$

$$W_{\text{col}} \in \mathbb{R}^{h \times h/4}. \tag{50}$$

One embedding is generated for each of the $k$ columns and concatenated together. A new positional embedding is generated for each forward pass, meaning the positional embeddings are randomised, for both training and inference. The targets $y^{\text{fit}}$ and $y^*$ are embedded with a linear transform,

$$h^y = \text{Linear}_y(y || \text{CDF}(y) || I_{\text{type}}) \in \mathbb{R}^{n \times h} \qquad y \in \{y^{\text{fit}}, y^*\} \tag{51}$$

where $I_{\text{type}} \in \{0, 1, 2\}$ is an indicator for the type of $y$, with value 0 for $y^{\text{fit}}$, 1 for counterfactual queries where $y^*$ is observed, and 2 for counterfactual queries where $y^*$ is unobserved. The CDF embedding is used here again. Next, we embed the treatment variables,

$$h^t = \text{emb}_t(t^{\text{fit}}) \in \mathbb{R}^{n \times 1 \times h} \tag{52}$$

$$h^{t*} = \text{emb}_{t*}(t^*) \in \mathbb{R}^{1 \times 1 \times h}. \tag{53}$$

Note the embedding for $t^*$ has 4 values corresponding to if the query is interventional or observational. These are concatenated to generate the full input embeddings,

$$h_0^{\text{fit}} = \text{Concat}(h^t, h^x, h^y, \text{dim} = 1) \in \mathbb{R}^{n \times (k+2) \times h} \tag{54}$$

$$h_0^{\text{pred}} = \text{Concat}(h^{t*}, h^{x*}, \mathbf{0}, \text{dim} = 1) \in \mathbb{R}^{1 \times (k+2) \times h}. \tag{55}$$

Finally, the prior ADMG graph structure is embedded. The ADMG is represented as a directed adjacency matrix, $A$, and bidirected correlation matrix $C$. Both matrices are binary. If the matrices are unknown, they are filled with zeros. From $A$, we create the ancestral matrix $R$, the matrix consisting of all ancestors for each node (reachability or transitive closure). The graph embedding is

$$h^G = \text{emb}_A(A) + \text{emb}_R(R) + \text{emb}_C(C) + \text{emb}_{AT}(A^T) + \text{emb}_{RT}(R^T) + \text{emb}_{CT}(C^T). \tag{56}$$

where $\text{emb}()$ is an elementwise learned binary embedding. This representation is heavily redundant since the transpose matrices and ancestral matrices are constructed from $A$ and $C$. This approach allows the model to easily identify every node's parents and children as well as ancestors and descendants without multiple hops.

Unlike TabPFNv2 (and Do-PFN), we omit input augmentations: grouping features, ensembling, random augmentations and random feature products. As well as simplifying the model, this is also required since the structure is explicitly represented in the input, so mixing features would not easily work. However, the CDF augmentation is kept in our model, as a fixed part of the encoder.

## E.2. Observation module

The observation module processes $h_0^{\text{fit}}$. The embeddings are passed through $L$ tabular transformer layers with self attention. Each transformer block consists of a row-wise attention, column-wise attention and a feed forward layer.

$$h_{\text{row}} = \text{MultiHeadAttention}(h, dim = 1) \tag{57}$$

$$h_{\text{col}} = \text{MultiHeadAttention}(h, dim = 0) \tag{58}$$

$$h_{\text{mlp}} = \text{FFN}(h) \tag{59}$$

$$h_l^{\text{fit}} = \text{Block}(h_{l-1}^{\text{fit}}) \tag{60}$$

where Block denotes a pre-layer-normalized transformer block with residual sublayers values using $h_{\text{row}}$, $h_{\text{col}}$, and $h_{\text{mlp}}$ and $h$ is the post layer normalised residual input for each block.

Let $h_{l,i}^{\text{fit}}$ be the output of layer $l$, feature $i$. The matrix decoder module uses the final embeddings $h_{L,i}^{\text{fit}}$ to generate predicted graph structure matrices. The embeddings are averaged over fit rows and projected using two linear layers with RMS normalisation to map each output to embeddings $u_i, v_i \in \mathcal{R}^h$. The logit for each edge is

$$\hat{M}_{i,j} = \alpha \cdot \text{gelu}(u_i^T \cdot W) \cdot v_j + \beta, \qquad \hat{M} \in \{\hat{A}, \hat{R}, \hat{C}\} \tag{61}$$

with learned parameters $W \in \mathcal{R}^{d \times d}$ and scalar $\alpha$ and $\beta$. Separate decoder parameters are used for each output matrix, $\hat{A}, \hat{R}, \hat{C}$. Entries in $\hat{M} \in \mathcal{R}^{n \times n}$ are logits representing the probability of each edge occurring. Empirically, we found this nonlinear decoder worked better than a direct dot product used in AVICI (Lorch et al., 2022). The matrix decoder's predicted graph depends only on $\mathcal{D}^{\text{fit}}$, and not on $\mathcal{D}^{\text{pred}}$ or the input graph structure.

## E.3. Graph module

The graph module takes the graph embeddings and the intermediate hidden states from the observation module to generate graph hidden states. These hidden states use feature observation hidden states for both positional embeddings and to pass information. The intermediate embeddings $h_l^{\text{col}}$ are averaged over the column dimension to create feature embeddings, $\bar{h}_l^{\text{col}} \in \mathcal{R}^{(k+2) \times h}$. These embeddings are then transformed into source and destination embeddings using two separate linear layers. The resulting representations are combined via an outer product to produce the feature matrix $H_l^{\text{feat}} \in \mathcal{R}^{(k+2) \times (k+2) \times h}$. The graph hidden states is computed as

$$H_l^G = \text{LN}(h^G + H_l^{\text{feat}}). \tag{62}$$

Note that this module only depends on $\mathcal{D}^{\text{fit}}$ and $G$, not on $\mathbf{x}^*$. Also, $h^G$ is not updated, we always use the initial $h^G$ from the encoder module. Empirical testing found no improvements from updating these weights between each layer.

### E.4. Prediction module

The prediction module generates the distribution $\hat{p}_\theta(y)$, using the previous embeddings and encoded prediction inputs. This module follows the same architecture and parameterization as the observation module, with the addition of a cross-attention layer over the graph module embeddings:

$$h^*_{\text{row}} = \text{MultiHeadAttention}(h^*, \dim = 1), \tag{63}$$

$$h^*_{\text{col}} = \text{MultiHeadAttention}(\text{KV} = h_{\text{row}}, \text{Q} = h^*, \dim = 0), \tag{64}$$

$$h^*_G = \text{MultiHeadAttention}(\text{KV} = H^G, \text{Q} = h^*, \dim = 0), \tag{65}$$

$$h_{\text{mlp}} = \text{FFN}(h^*_G), \tag{66}$$

$$h^*_l = \text{Block}(h^*_{l-1}). \tag{67}$$

Block is a pre-layer norm residual block using $h^*_{\text{row}}, h^*_{\text{col}}, h_{\text{mlp}}$ residual sublayers, with residual stream $h^*$. Cross attention to the observation embeddings $h_{\text{row}}$ and $H^G$ allows the prediction module to access the observational data and graph structure. Finally, the predicted outcome is generated from the final embedding corresponding to the outcome feature, $h^*_{L,y} \in \mathcal{R}^h$ using a MLP head, mapping to prediction bucket logits,

$$z = \text{MLP}(h^*_{L,y}) \in \mathbb{R}^b \tag{68}$$

$$\hat{p}_\theta(y) = \text{softmax}(z) \in \mathbb{R}^b \tag{69}$$

where $z$ are output logits, $b$ is the number of buckets in the discretized output distribution and $\hat{p}_\theta(y)$ is the predicted probability distribution over these buckets.

### E.5. Architecture changes

Several additional architecture optimisations and training improvements were made compared to the transformer backbone used in TabPFNv2. These improvements are aimed at improving training stability and efficiency, rather than adding specific model features. Changes are largely inspired by the "Modded NanoGPT" project (Jordan et al., 2024a). The main changes found to work are:

- Full attention to $h_{\text{row}}$ in the prediction module.

- Cosine learning rate schedule with 1000 linear warmup steps.

- relu$^2$ activation instead of relu/gelu (Zhang et al., 2024).

- Reduce positional embedding initialisation weight scale.

- Pre layer norm instead of post layer norm (Xiong et al., 2020).

- Apply layer norm before prediction decoder MLP head.

- Muon optimizer for matrices instead of AdamW (Jordan et al., 2024b), using the Kimi variant (Liu et al., 2025).

- QK norm with learnable temp for attention heads (Henry et al., 2020).

- Increased width from 192 to 288. Total parameters increase from 10.28M to 22.97M.

We test the effect of these changes in the ablation section F, and find they significantly improve training stability and efficiency.

## F. Architecture Ablations

This section conducts an ablation of the architecture changes described in Section E.5. The TabPFN-CFM architecture consists of a modified Do-PFN architecture to handle the additional structural and outcome prediction objectives. Independently, we introduce multiple updates to the transformer backbone and training procedure to improve training efficiency and final performance. No new functionality is added from these changes.

To evaluate the impact of each of the changes, we conduct an ablation study on on smaller datasets with models trained for 30k steps. The training and validation datasets are identical on all setups to isolate the effect of architecture changes. The learning rate is 3e-4 for Muon and 1e-4 for AdamW, as AdamW required a lower learning rate for stability. Results are shown in Figure 3. We sequentially apply each of the architectural changes on top of previous changes and record the prediction and adjacency losses. Both prediction and adjacency losses decrease with the changes. Note changes interact with each other in nonlinear ways and some changes only show in longer training runs, so the impact of each change in the final model is more complex than shown in the plots here.

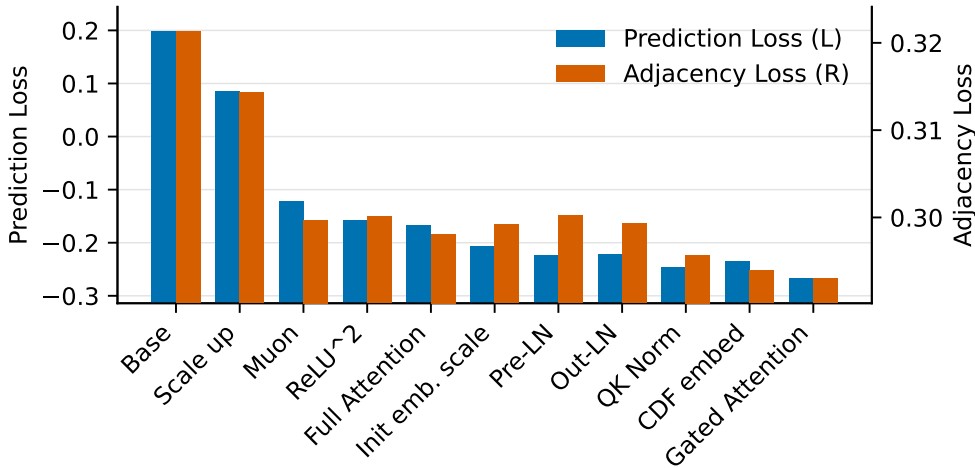

*Figure 3.* Ablation results with 30k training steps for prediction loss and adjacency matrix loss. Changes are applied sequentially from left to right.

Next, we compare our model with longer training runs with and without the changes, but with the base version scaled up to the same 23.2M parameter count for fairness. Models are trained for a much longer 150k steps. Figure 4 shows validation loss curves for both models. Our optimised model reaches the same prediction loss 4 times faster (37K vs 150K steps to reach -0.195 loss) and 3 times faster adjacency loss (50K vs 150K steps to reach 0.283 loss) compared to the unoptimised version, with significantly lower final loss. The base model trained in 8.52 hours while the final model trained in 10.24 hours, a 20% increase in training time largely from Muon and the additional norm layers. Despite a small increase in per step time, the final model is significantly more efficient.

# G. Synthetic toy examples

### G.1. Linear Instrumental Variable example

Showcase experiment on the Instrumental Variable (IV) problem, with SEM: $Z \rightarrow T \rightarrow Y, U \rightarrow T, U \rightarrow Y$. To demonstrate, we examine predictions from a simple random synthetic linear SEM. The SEM equations (after normalizing with the observational sample statistics) are:

$$
\begin{aligned}
\epsilon_{\{u,z,t,y\}} &\sim U(-0.4, 0.4) \\
U &= 4.303\epsilon_y - 0.031 \\
Z &= 4.380\epsilon_z - 0.007 \\
T &= \text{binarize}_{0.5}(-0.105z + 0.104u + \epsilon_t) \\
Y &= -1.818 * T - 0.108 * U + 1.047 * \epsilon_y + 0.906
\end{aligned}
$$

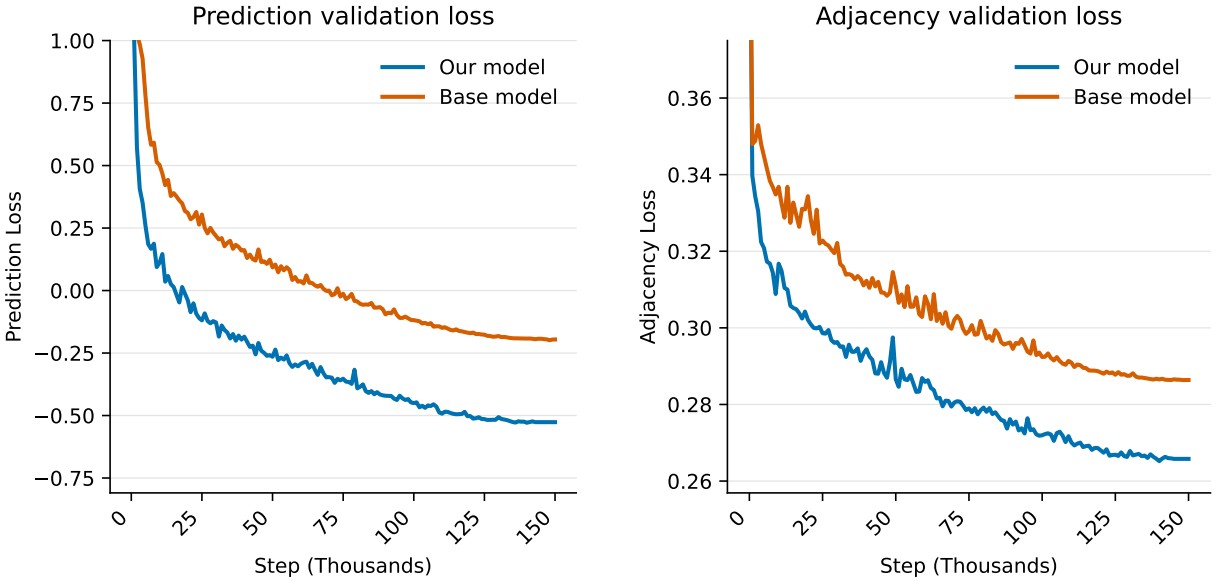

*Figure 4.* Loss curves comparing our and baseline architecture for 150k training steps, prediction loss (left), adjacency matrix loss (right).

where the binarize function returns 1 if the input is in the top 50 percent of the overall distribution and 0 otherwise. We draw a single sample from this SEM:

$$z = -0.187$$
$$u = 0.421$$
$$t = 1$$
$$y = -0.568$$

Now, we identify the exact distribution assuming everything is observed, for observational, interventional, and counterfactual queries. The observational distribution of $y$ is

$$Y = -0.957 + 1.047 * \epsilon_y,$$

the interventional distribution with $do(T = 0)$ is

$$Y = 0.861 + 1.047 * \epsilon_y$$

and the counterfactual distribution, which assumes $y$ is observed, is

$$Y = 1.250$$

since $\epsilon_y = 0.372$ can be determined by the observation of $y$, which allows for the exact counterfactual value to be computed with no uncertainty.

### G.2. Nonlinear SEM

A more complex nonlinear SEM is used to test the model. The graph structure is fixed as $Z \to T \to V \to Y, U \to V, U \to W \to Y$ where $U$ is unobserved. The graph contains latent variables and covariates that are descendants of $T$. The SEM equations are drawn from our nonlinear prior distribution. Since the exact posterior for interventional and observational distributions is intractable, we compare the model predictions to a single sample drawn from the true distribution. Results are shown for $n = 512$. Figure 6 shows the model predictions align well with the samples, even without the graph structure. Giving the graph structure improves predictions. Table 11 shows the true and predicted adjacency matrices. The predicted

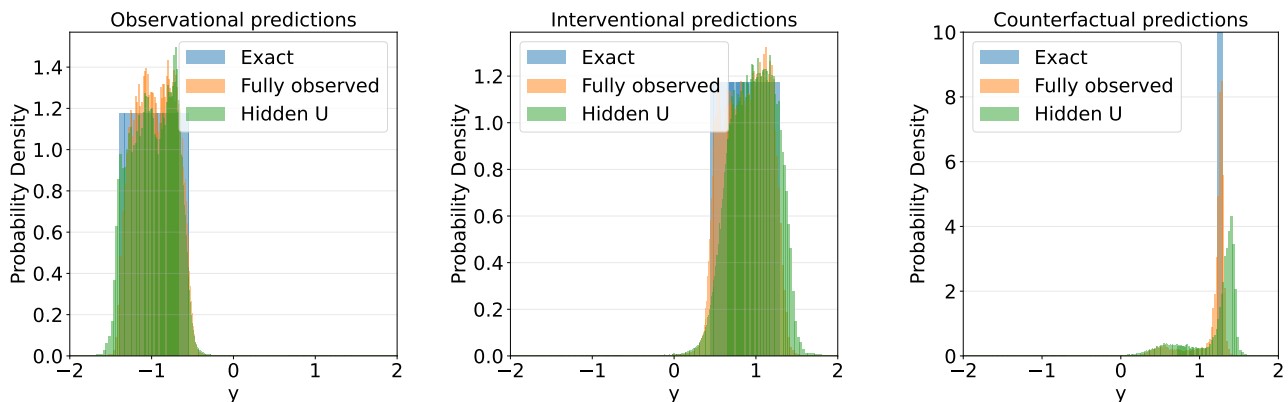

*Figure 5.* Interventional (left), Observational (center), and Counterfactual (right) distributions. Exact solutions are in blue, model predictions in orange for when $U$ is observed, and in green when $U$ is unobserved. Model is given the graph structure.

adjacency matrix generally matches the true adjacency matrix, though there is some uncertainty in the children of $T$ likely due to $V, W, Y$ all being correlated. In Figure 7, we show the predicted distribution shifts closer towards the true counterfactual as $n$ increases (without $\mathcal{G}$). For small $n$, the counterfactual is biased towards the observed $y^*$, which vanishes as $n$ increases and the model is able to learn the underlying SEM better.

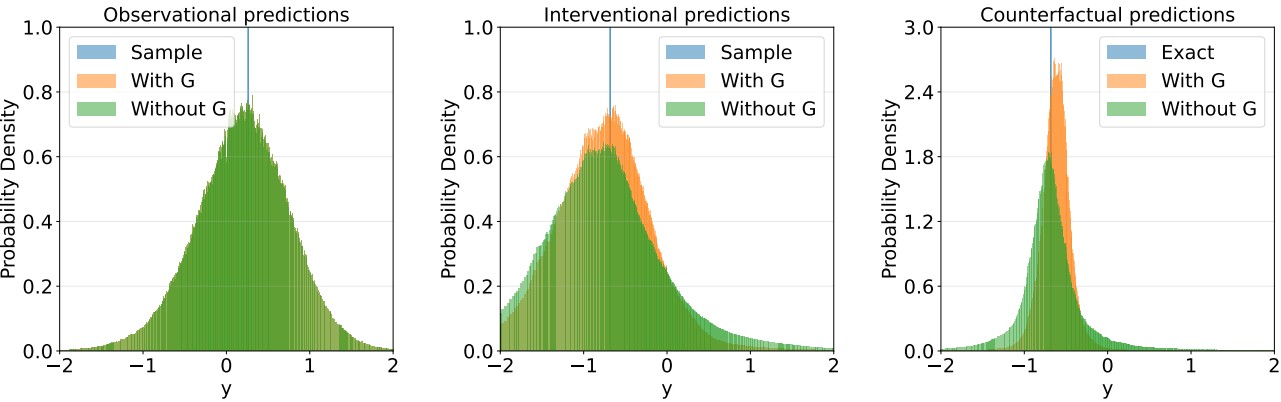

*Figure 6.* Interventional (left), Observational (center), and Counterfactual (right) distributions. Predictions are compared with and without the true graph structure $\mathcal{G}$. The Observational and Interventional predictions are compared to a single sample drawn from the true distribution, while the Counterfactual predictions are compared to the true counterfactual value.

## H. OOD data generation

We test our model on datasets drawn using random Fourier functions as nonlinearities. This has been used as part of the prior in other works, but our models are not trained on these functions, so they are OOD. The distribution of SEM functions is constructed as follows. First, parameters are sampled,

$$w \sim U[0.1, 0.5] \tag{70}$$
$$c \sim U[1.41, 5.65] \tag{71}$$
$$b \sim U[-0.5, 0.5] \tag{72}$$
$$W_{i,j} \sim \text{Cauchy}(0, w) \tag{73}$$
$$\phi_i \sim N(0, 2\pi) \tag{74}$$
$$k_i \sim N(0, c(n_{\text{in}})^{-0.5}). \tag{75}$$

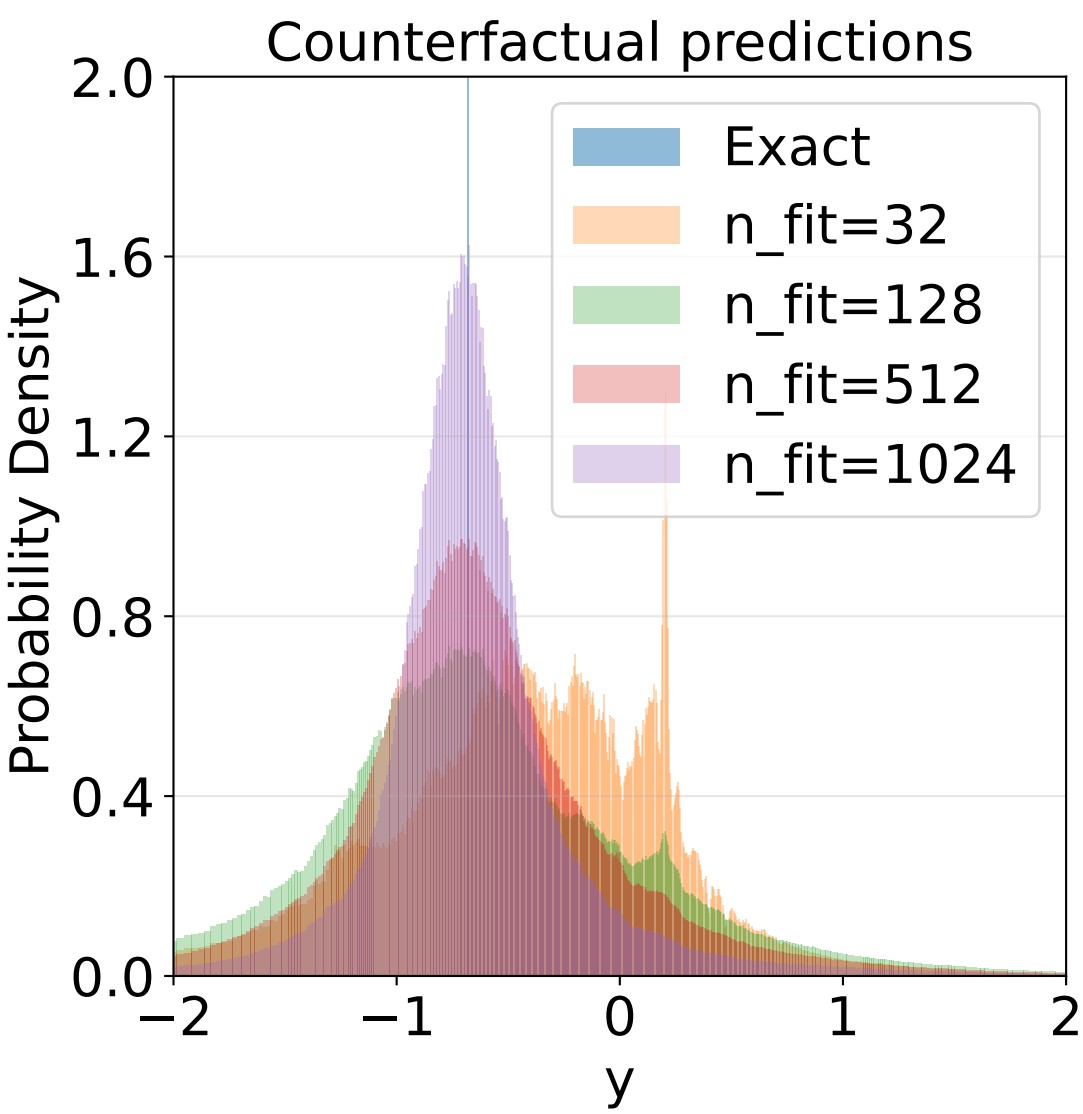

*Figure 7.* Counterfactual distribution predictions for the nonlinear SEM with different $\mathcal{D}^{\text{fit}}$ sample sizes.

*Table 11.* True and predicted adjacency matrices for the nonlinear SEM experiment. Left: True adjacency matrix. Right: Predicted adjacency matrix (values represent predicted edge probabilities).

|   | T | Z | V | W | Y |   |   | T | Z | V | W | Y |
|---|---|---|---|---|---|---|---|---|---|---|---|---|
| **T** | 0 | 0 | 1 | 0 | 0 |   | **T** | 0.000 | 0.002 | 0.992 | 0.938 | 0.552 |
| **Z** | 1 | 0 | 0 | 0 | 0 |   | **Z** | 0.982 | 0.000 | 0.318 | 0.522 | 0.391 |
| **V** | 0 | 0 | 0 | 0 | 1 |   | **V** | 0.000 | 0.000 | 0.000 | 0.597 | 0.758 |
| **W** | 0 | 0 | 0 | 0 | 1 |   | **W** | 0.006 | 0.005 | 0.573 | 0.000 | 0.871 |
| **Y** | 0 | 0 | 0 | 0 | 0 |   | **Y** | 0.000 | 0.000 | 0.001 | 0.005 | 0.000 |

*Table 12.* Predictions on in distribution SEMs with $n = 1024$ fit samples, compared by mean squared error ($\downarrow$). The baseline methods are unable to make counterfactual predictions.

| Model | Obs. | Int. | Count. |
|---|---|---|---|
| S-learner | 0.430 | 0.797 | - |
| T-learner | 0.434 | 0.846 | - |
| X-learner | 0.434 | 0.846 | - |
| DR-learner | 0.431 | 0.825 | - |
| Do-PFN | 0.670 | 0.941 | - |
| TabPFN-CFM | 0.433 | 0.578 | 0.247 |
| TabPFN-CFM W/Graph | 0.433 | 0.551 | 0.215 |

The output of each node is

$$y = b + \sum_{i=1}^{32} k_i \cos\big((xW^\top)_i + \phi_i\big) \tag{76}$$

where $x$ is a concatenation of the node's parent values and noise variables, and parameters $W, k, \phi$ are randomly sampled frequency, magnitude and phase for each of the 32 coefficients. This process generates random functions that exhibit non-trivial nonlinear behaviors over a range of frequencies.

*Table 13.* Structural predictions on in distribution SEMs with $n = 1024$ fit samples, evaluated on predicted adjacency matrix, ancestral matrix and confounding matrix. Main values tracks the AUROC ($\uparrow$), with accuracy ($\uparrow$)in brackets.

| Model | Adjacency | Ancestral | Confounding |
|---|---|---|---|
| FCI | 0.533 (0.89) | 0.536 (0.81) | 0.54 (0.90) |
| GES | 0.672 (0.81) | 0.657 (0.60) | - |
| LiNGAM | 0.568 (0.78) | 0.540 (0.70) | - |
| PC | 0.651 (0.84) | 0.632 (0.46) | - |
| AVICI | 0.703 (0.89) | 0.682 (0.82) | - |
| TabPFN-CFM | 0.902 (0.91) | 0.922 (0.88) | 0.877 (0.94) |

*Table 14.* Predictions on in distribution SEMs with varying fit samples (second header), compared by mean squared error (↓). The baseline methods are unable to make counterfactual predictions.

| Model | Observational | | | Interventional | | | Counterfactual | | |
|---|---|---|---|---|---|---|---|---|---|
| | 32 | 256 | 1024 | 32 | 256 | 1024 | 32 | 256 | 1024 |
| S-learner | 0.591 | 0.457 | 0.430 | 0.942 | 0.797 | 0.797 | - | - | - |
| T-learner | 0.645 | 0.466 | 0.434 | 1.209 | 0.852 | 0.846 | - | - | - |
| X-learner | 0.645 | 0.466 | 0.434 | 1.209 | 0.852 | 0.846 | - | - | - |
| DR-Learner | 0.645 | 0.461 | 0.431 | 0.987 | 0.807 | 0.825 | - | - | - |
| Do-PFN | 1.508 | 0.796 | 0.670 | 2.322 | 1.080 | 0.941 | - | - | - |
| TabPFN-CFM | 0.616 | 0.460 | 0.433 | 0.804 | 0.619 | 0.578 | 0.400 | 0.276 | 0.247 |
| TabPFN-CFM With Graph | 0.616 | 0.460 | 0.433 | 0.769 | 0.583 | 0.551 | 0.355 | 0.239 | 0.215 |

*Table 15.* Structural predictions on in distribution SEMs with varying fit samples. Main values track AUROC (↑), with accuracy (↑) in brackets.

| Model | Adjacency Matrix | | | Ancestral Matrix | | |
|---|---|---|---|---|---|---|
| | 32 | 256 | 1024 | 32 | 256 | 1024 |
| FCI | 0.523 (0.89) | 0.532 (0.89) | 0.533 (0.89) | 0.516 (0.82) | 0.521 (0.81) | 0.536 (0.81) |
| GES | 0.585 (0.82) | 0.637 (0.82) | 0.672 (0.81) | 0.616 (0.67) | 0.651 (0.63) | 0.657 (0.60) |
| LiNGAM | 0.530 (0.80) | 0.556 (0.79) | 0.568 (0.78) | 0.506 (0.72) | 0.524 (0.71) | 0.540 (0.70) |
| PC | 0.587 (0.87) | 0.632 (0.86) | 0.651 (0.84) | 0.580 (0.78) | 0.637 (0.56) | 0.632 (0.46) |
| AVICI | 0.611 (0.88) | 0.701 (0.89) | 0.703 (0.89) | 0.572 (0.80) | 0.683 (0.81) | 0.682 (0.82) |
| TabPFN-CFM | 0.774 (0.89) | 0.870 (0.90) | 0.902 (0.91) | 0.783 (0.82) | 0.887 (0.86) | 0.922 (0.88) |

| Model | Confounding Matrix | | |
|---|---|---|---|
| | 32 | 256 | 1024 |
| FCI | 0.533 (0.92) | 0.546 (0.91) | 0.540 (0.90) |
| GES | - | - | - |
| LiNGAM | - | - | - |
| PC | - | - | - |
| AVICI | - | - | - |
| TabPFN-CFM | 0.806 (0.94) | 0.858 (0.94) | 0.877 (0.94) |

*Table 16.* Predictions on out of distribution SEMs with varying fit samples (second header), compared by mean squared error (↓). The baseline methods are unable to make counterfactual predictions.

| Model | Observational | | | Interventional | | | Counterfactual | | |
|---|---|---|---|---|---|---|---|---|---|
| | 32 | 256 | 1024 | 32 | 256 | 1024 | 32 | 256 | 1024 |
| S-learner | 0.811 | 0.623 | 0.565 | 1.084 | 0.896 | 0.888 | - | - | - |
| T-learner | 0.864 | 0.642 | 0.574 | 1.283 | 0.949 | 0.909 | - | - | - |
| X-learner | 0.863 | 0.462 | 0.574 | 1.283 | 0.949 | 0.909 | - | - | - |
| DR-Learner | 0.850 | 0.648 | 0.577 | 1.104 | 0.896 | 0.885 | - | - | - |
| Do-PFN | 1.336 | 0.814 | 0.782 | 2.599 | 1.006 | 0.966 | - | - | - |
| TabPFN-CFM | 0.803 | 0.600 | 0.537 | 1.010 | 0.802 | 0.765 | 0.809 | 0.654 | 0.607 |
| TabPFN-CFM With Graph | 0.803 | 0.600 | 0.538 | 0.991 | 0.778 | 0.739 | 0.766 | 0.616 | 0.572 |

*Table 17.* Structural predictions on synthetic OOD SEMs with varying fit samples. Main values track AUROC (↑), with accuracy (↑) in brackets.

| Model | Adjacency Matrix | | | Ancestral Matrix | | |
|---|---|---|---|---|---|---|
| | 32 | 256 | 1024 | 32 | 256 | 1024 |
| FCI | 0.517 (0.88) | 0.524 (0.88) | 0.533 (0.88) | 0.512 (0.82) | 0.515 (0.81) | 0.523 (0.81) |
| GES | 0.550 (0.83) | 0.609 (0.83) | 0.654 (0.81) | 0.554 (0.73) | 0.616 (0.67) | 0.654 (0.60) |
| LiNGAMICA | 0.516 (0.81) | 0.544 (0.81) | 0.558 (0.78) | 0.501 (0.74) | 0.520 (0.73) | 0.524 (0.70) |
| PC | 0.587 (0.87) | 0.656 (0.87) | 0.685 (0.86) | 0.578 (0.78) | 0.628 (0.62) | 0.639 (0.55) |
| AVICI | 0.618 (0.88) | 0.637 (0.89) | 0.623 (0.89) | 0.593 (0.81) | 0.619 (0.82) | 0.596 (0.82) |
| TabPFN-CFM | 0.716 (0.89) | 0.837 (0.90) | 0.889 (0.91) | 0.712 (0.83) | 0.848 (0.85) | 0.902 (0.88) |

| Model | Confounding Matrix | | |
|---|---|---|---|
| | 32 | 256 | 1024 |
| FCI | 0.532 (0.90) | 0.552 (0.89) | 0.563 (0.87) |
| GES | - | - | - |
| LiNGAMICA | - | - | - |
| PC | - | - | - |
| AVICI | - | - | - |
| TabPFN-CFM | 0.757 (0.92) | 0.807 (0.93) | 0.828 (0.93) |

