# OpenReview forum: "A Causal Foundation Model for Structure and Outcome Prediction"
_ICML.cc/2026/Workshop/FMSD — FMSD @ ICML 2026 Poster_

### Official Review · Reviewer_iFnM · 2026-05-19
**TabPFN-CFM**

**Rating:** 7
**Confidence:** 4

**Review:**

## Summary

This paper introduces TabPFN-CFM, a causal foundation model that jointly handles (1) outcome prediction across all three levels of Pearl's Causal Hierarchy (observational, interventional, counterfactual) and (2) causal structure prediction, including unobserved confounding via Acyclic Directed Mixed Graphs (ADMGs).

## Strengths

- **Ambitious unification**: Joint training for structure and outcome prediction across all three rungs of Pearl's hierarchy is a substantive extension of prior PFN-based causal models. The motivation that joint training yields broader causal understanding is well-articulated.
- **Theoretical justification for graph conditioning**: The KL-divergence-based proof in Appendix A that conditioning on the graph cannot hurt posterior estimation is clean and correct, with an explicit characterization of when no improvement occurs.
- **ADMG handling**: Explicit representation of bidirected (confounding) edges is a meaningful extension over methods that assume only DAGs. The benchmarks show strong AUROC on confounding prediction (0.828 OOD), which baseline methods do not even attempt.
- **Training efficiency**: The architectural and optimization changes (Muon, ReLU², QK norm, pre-LN, etc.) yield ~3x training speedup with lower final loss (Figure 4). This is a practical contribution that benefits downstream researchers.
- **Strong synthetic results**: TabPFN-CFM consistently outperforms Do-PFN and meta-learner baselines on interventional and counterfactual tasks (Tables 1, 11). Structure prediction substantially outperforms classical methods (FCI, GES, LiNGAM, PC) and AVICI (Table 2).
- **Generalization to OOD SEMs**: Performance on random-Fourier-feature SEMs (not seen in training) is preserved, suggesting the model isn't overfitting to its prior.
- **Real-data validation**: The Amazon Sales and Law School Admissions evaluations (Tables 4-5) demonstrate the model isn't purely a synthetic-data artifact.


## Areas for Improvement

- **Compression hurts clarity**: The workshop format forces a lot into the appendices. The main paper barely sketches the architecture, with the bulk of detail in Appendix E. Figure 1 is information-dense but hard to parse without the appendix. Some readers may not be able to follow the contributions from the main text alone.
- **Binary-treatment restriction is not flagged early enough**: The treatment $T \in \{0, 1\}$ assumption is stated in Section 1.1 but the breadth of the framing ("multiple causal problems," "all three levels of Pearl's hierarchy") could mislead readers into thinking continuous treatments are handled.
- **Real-data graph predictions are mixed**: Table 7 shows AUROC 0.886 on Law School adjacency but Table 6 shows only 0.708 on Amazon Sales adjacency. The text doesn't address why structure prediction quality varies so much across the two real datasets, which is important information for practitioners.
- **Architecture changes are bundled**: The ablation in Figure 3 adds changes sequentially, so the marginal impact of each is muddled by interactions. Some changes (Muon, scale-up) likely drive most gains, but this is hard to disentangle.
- **No comparison to CausalPFN**: CausalPFN is cited as foundational, but no head-to-head comparison on its evaluation benchmarks is included. Since CausalPFN handles continuous-valued outcomes for binary-treatment ATE estimation, a comparison would clarify positioning.
- **"3x speedup" claim should be on optimization-equivalent runs**: Figure 4 compares the modified architecture and base architecture, but with different effective hyperparameters (width 192 → 288 doubles parameters). The speedup claim mixes algorithmic and capacity gains.
- **Limited interpretability/diagnostic analysis**: For a model handling confounding, examples showing the model identifying genuine confounders in real data would strengthen the contribution beyond aggregate metrics.

## Detailed Comments

1. The dummy features from batch padding "may support intermediate computations". This is intriguing but speculative. Did you investigate what features dummy tokens learn? Probing analysis would be valuable.
2. The graph posterior $P(G | D^{\text{obs}})$ depends only on $D^{\text{fit}}$, not on $D^{\text{pred}}$ (Section 3.1). What happens when the prediction query violates assumptions in the observational data (e.g., out-of-distribution covariates)? Does the predicted graph adapt at all?
3. The cross-attention from prediction module to graph module is one-directional. Have you investigated whether bidirectional graph-prediction interaction would help, perhaps at increased compute cost?
4. The discussion section is very short. What are the failure modes? When should practitioners not use TabPFN-CFM?

## Justification of Score

This is a technically strong paper with ambitious scope (joint structure + outcomes + all 3 Pearl rungs + ADMGs) and a clear empirical demonstration of value. The architectural and training innovations are practically useful. However, the binary-treatment restriction limits some claims, the presentation is dense, and several baselines are missing (CausalPFN, head-to-head with continuous-treatment models). The contribution to the workshop is clear and substantive.

---

### Official Review · Reviewer_mUXC · 2026-05-22
**An interesting causal foundation model that can handle multiple causal problems.**

**Rating:** 7
**Confidence:** 3

**Review:**

This paper proposes TabPFN-CFM, a causal foundation model that can handle multiple causal problems. The model extends prior PFN-based causal inference approaches by supporting all three levels of Pearl’s causal hierarchy and incorporating graph priors when available. Similar to TabPFN, TabPFN-CFM is trained on synthetic datasets, and generalizes to real datasets, and shows good performance.

Pros:
1. Good empirical results on structure learning.
2. Practical architecture improvements.

Cons:
1. The real-world evaluation is limited. Although this is common in the study of causal inference, validation based on real-world datasets would make the empirical studies more convincing.
2. The concern of scalability remains.